# Digital Educational Intervention to Improve Adherence and Self-Care in Chronic Patients: A Prospective Study Protocol (PROSELF)

**DOI:** 10.3390/healthcare13222972

**Published:** 2025-11-19

**Authors:** Angelo Cianciulli, Giovanni Boccia, Roberta Manente, Antonietta Pacifico, Giuseppina Speziga, Emanuela Santoro

**Affiliations:** 1Department of Medicine, Surgery and Dentistry “Scuola Medica Salernitana”, University of Salerno, 84081 Baronissi, Italy; gboccia@unisa.it (G.B.); apacifico@unisa.it (A.P.); gspeziga@unisa.it (G.S.); esantoro@unisa.it (E.S.); 2Integrated Care Department of Health Hygiene and Evaluative Medicine, San Giovanni di Dio e Ruggi d’Aragona University Hospital, Largo Città di Ippocrate, 1, 84131 Salerno, Italy; 3Hospital and Epidemiological Hygiene Unit, San Giovanni di Dio e Ruggi d’Aragona University Hospital, 18 Hospital, Largo Città di Ippocrate, 1, 84131 Salerno, Italy; 4San Giovanni di Dio e Ruggi d’Aragona University Hospital, 84081 Salerno, Italy; manente392@gmail.com

**Keywords:** self-care, adherence, COPD, diabetes, cardiovascular disease, digital health, nurse-led education, tele-education, chronic diseases, patient education, study protocol, community nursing, PNRR Mission 6

## Abstract

**Background:** Chronic non-communicable diseases—chiefly chronic obstructive pulmonary disease (COPD), type 2 diabetes mellitus (T2DM), and cardiovascular diseases (CVDs)—remain the leading causes of morbidity and mortality worldwide. Despite the proliferation of telehealth programs, few longitudinal studies have rigorously evaluated theory-based, nurse-led digital education across multiple chronic conditions within a unified self-care framework. PROSELF (Promotion of Self-Care through Educational Interventions in Chronic Patients) addresses this gap. **Methods:** Promotion of Self-Care (PROSELF) is a prospective, longitudinal, multicenter, non-interventional study in community and primary-care settings in Southern Italy. A 12-month nurse-led digital educational intervention—grounded in the Middle-Range Theory of Self-Care of Chronic Illness—targets the three core self-care dimensions (maintenance, monitoring, and management) through individualized tele-education, asynchronous reinforcement, and structured follow-up at baseline, 3, 6, and 12 months. Validated self-care, quality-of-life, and social-support measures will be used. Data will be analyzed using repeated-measures and multivariate models to evaluate longitudinal changes in self-care, adherence, and related outcomes. Participation requires digitally informed consent. **Results:** The PROSELF study is expected to demonstrate the effectiveness of a 12-month, nurse-led digital educational program in improving adherence and self-care behaviors among patients with chronic diseases (COPD, diabetes, and cardiovascular disease). The intervention will leverage validated self-care assessment tools and tele-educational follow-up sessions. **Conclusions:** Findings from this study will inform the design of scalable, evidence-based, behaviorally informed models for digital chronic care delivery and nursing education.

## 1. Introduction

Chronic diseases account for approximately 41 million deaths annually and more than 70% of healthcare expenditure globally, placing sustained pressure on health systems and communities [1]. Pulmonary disease (COPD), type 2 diabetes mellitus (T2DM), and cardiovascular diseases (CVDs) share a long-term course, high risk of exacerbations, and a strong dependence on daily behaviors—adherence to therapy, self-monitoring, and timely help-seeking that cannot be achieved by clinical encounters alone [2,3,4]. In this landscape, self-care has emerged as a cornerstone of person-centered chronic care.

The growing burden of chronic diseases has exposed critical weaknesses in traditional healthcare models, which often rely on episodic, hospital-centered care and fail to address patients’ long-term behavioral needs. In this context, the integration of digital health technologies and nurse-led educational interventions has emerged as a promising approach to improve self-management and continuity of care. However, despite the growing availability of digital tools, structured and theory-based educational models led by nurses remain scarce, particularly in community and primary care settings.

The necessity of the Promotion of Self-Care through Educational Interventions in Chronic Patients (PROSELF) protocol arises from this gap. It aims to operationalize the goals of the Italian National Recovery and Resilience Plan (PNRR—Mission 6 “Health”), which emphasizes proximity care, patient empowerment, and the strategic role of Family and Community Nurses (FCNs) in digitally supported chronic disease management. By designing and evaluating a 12-month, nurse-led digital educational program grounded in the Middle-Range Theory of Self-Care of Chronic Illness, PROSELF seeks to provide a reproducible framework that strengthens evidence-based practice and informs future policy development in community health nursing.

The Middle-Range Theory of Self-Care of Chronic Illness defines self-care as a cyclical, adaptive process comprising maintenance (health-promoting and adherence behaviors), monitoring (recognition/interpretation of signs and symptoms), and management (appropriate responses and decision-making in collaboration with professionals) [5]. Yet, despite decades of initiatives, adherence and self-management remain suboptimal due to low health literacy, variable digital competence, and fragmentation across care pathways [6,7,8].

The COVID-19 pandemic accelerated the adoption of telehealth and digital education, revealing their potential to sustain continuity of care and support behavioral change at scale [9,10]. International experiences—particularly in the UK and Scandinavian countries—show that community nurses can effectively lead digital self-management programs, improving patient engagement and outcomes [11,12]. In Italy, the Piano Nazionale di Ripresa e Resilienza (PNRR)—Missione 6 “Salute” places the Family and Community Nurse (FCN) at the center of territorial, digitally supported chronic care. However, robust longitudinal evaluations of theory-driven, nurse-led digital education within this new framework are lacking [13,14].

PROSELF is designed to fill this gap. The intervention is explicitly theory-based [5], complemented by behavioral frameworks—Bandura’s Self-Efficacy and the Health Belief Model—to strengthen motivational components and perceived benefits/barriers [5,15,16].

Both theoretical frameworks were selected for their complementary relevance to behavioral change in chronic disease self-management. Bandura’s Self-Efficacy Theory emphasizes that individuals’ belief in their capability to execute specific behaviors directly influences persistence, motivation, and long-term adherence—constructs particularly relevant to medication routines, symptom monitoring, and help-seeking behaviors. The Health Belief Model (HBM), on the other hand, explains how perceived risk, benefits, and barriers influence health decisions and engagement in preventive actions. Within the PROSELF framework, self-efficacy components such as mastery experience, feedback, and goal setting are applied uniformly across conditions, whereas HBM-related constructs (perceived severity, susceptibility, and cues to action) are emphasized mainly for cardiovascular and metabolic diseases, where risk perception has a stronger motivational value. This integration ensures that the intervention remains both theoretically grounded and adaptable to disease-specific behavioral determinants.

The study evaluates 12-month effects on global and domain-specific self-care and on secondary outcomes (adherence proxies, health-related quality of life, perceived social support), while identifying predictors of improvement.

The primary objective is to quantify changes in overall self-care at 12 months.

Secondary objectives are (i) to assess changes across maintenance, monitoring, and management domains; (ii) to evaluate the effects of nurse-led digital education on treatment adherence, health-related quality of life, and perceived social support; and (iii) to identify predictors of improvement and patient engagement during follow-up.

Research hypotheses are integrated with these objectives. Specifically, we hypothesize that

(1)Participation in the PROSELF program will significantly improve self-care maintenance, monitoring, and management behaviors over time (linked to the primary objective i);(2)Nurse-led digital educational interventions will enhance adherence and perceived self-efficacy (linked to secondary objective ii);(3)Improvements in self-care and adherence will be associated with a higher quality of life and stronger perceived social support at 12 months (linked to secondary objective iii).

These hypotheses will be tested using validated self-care, quality-of-life, and social-support measures, as detailed in Section 2.

## 2. Materials and Methods

### 2.1. Study Design

PROSELF is a prospective, longitudinal, multicenter, non-interventional protocol with four assessments: baseline (T0), 3 months (T1), 6 months (T2), and 12 months (T3). The design follows SPIRIT 2013 for protocol transparency and STROBE recommendations for observational studies [17,18]. The study aligns with the organizational and digital priorities of Italy’s National Recovery and Resilience Plan Mission 6—health.

### 2.2. Setting and Participants

The study will be conducted in community and primary-care services across Southern Italy, including FCN clinics, general practice networks, and chronic disease outpatient pathways. Inclusion: adults (≥18 years) with a confirmed diagnosis of COPD, T2DM, or CVD; clinically stable at enrollment; access to internet-enabled devices; ability to consent and participate in tele-education. Exclusion: cognitive or psychiatric conditions precluding participation—such as moderate to severe dementia, major depressive disorder with psychotic features, schizophrenia spectrum disorders, or acute manic or psychotic episodes—severe uncorrected sensory impairments, or concurrent enrollment in interventional trials. Participants will be referred by FCNs and General Practitioners (GPs) during routine care or teleconsultation. Additionally, participants who already demonstrate optimal or near-optimal self-care behaviors—such as consistent treatment adherence, regular symptom monitoring, and proactive disease management—will be excluded based on baseline screening, to ensure measurable intervention effects. Eligibility will be verified using data from participants’ electronic medical records, reviewed and confirmed by the referring Family and Community Nurses (FCNs) or General Practitioners (GPs). Only objectively documented diagnoses (COPD, T2DM, or CVD) and clinical stability will be considered valid for inclusion. Self-reported data will be limited to sociodemographic and behavioral variables such as education, smoking habits, and physical activity.

### 2.3. Sample Size Justification

Using G*Power 3.1.9.7. (Heinrich-Heine-Universität Düsseldorf, Düsseldorf, Germany), we powered a repeated-measures ANOVA (within subjects, 4 timepoints). With effect size f = 0.25 (≈Cohen’s d = 0.5, moderate), α = 0.05, power = 0.80, and correlation among repeated measures r = 0.50, the minimum required sample is N = 159. Allowing for 15% attrition, we plan to recruit N = 180 (≈60 per diagnostic group), ensuring adequate precision for primary and secondary analyses.

The selected sample size (N = 180) ensures adequate precision for detecting moderate within-subject effects (Cohen’s d ≈ 0.5) across four timepoints, representing a balance between feasibility and statistical power. This sample is consistent with or larger than comparable nurse-led self-care intervention studies in chronic patients (typically N = 100–150) [19,20,21]. Given the multicenter design and inclusion of three major chronic conditions (COPD, diabetes, and CVD), the findings will allow reasonable generalization to similar community and primary-care populations within Southern Italy and comparable European settings.

### 2.4. Training and Intervention Fidelity

All nurse interventors will complete a standardized 4 h training covering the following: (i) theory-driven education anchored to SC-CII domains; (ii) motivational interviewing techniques; (iii) telecommunication standards and privacy; (iv) digital health literacy coaching; and (v) documentation on the electronic case report form (eCRF). To ensure fidelity, a random 10% of tele-education sessions will be audited by a clinical supervisor using a structured checklist (adherence to content, dose, pacing, patient engagement, and shared goal setting). Deviations will prompt feedback and corrective training.

To ensure theoretical coherence and transparency, the intervention content was mapped to the Behaviour Change Techniques Taxonomy version 1 (BCTTv1) developed by Michie et al. [22]. Each educational session incorporates specific BCTs aligned with the self-efficacy and health belief frameworks, including “goal setting (behavior),” “self-monitoring of behavior,” “feedback on performance,” “problem solving,” “information about health consequences,” and “action planning.” This mapping allows for structured replication, facilitates fidelity assessment, and provides a framework for analyzing which components most effectively promote self-care maintenance, monitoring, and management behaviors.

### 2.5. Intervention

The PROSELF intervention is a nurse-led, theory-based digital program delivered via encrypted video calls and supported by asynchronous materials (brief videos, PDFs, checklists, reminders). Each session lasts 30–40 min and is individualized through shared agenda setting and goal planning. Content maps directly onto maintenance, monitoring, and management, integrating Bandura’s Self-Efficacy (mastery, vicarious experiences, persuasion, and emotional regulation) and Health Belief Model constructs (perceived risk, benefits, barriers, and cues to action).

Session 1: (T0, baseline; focus: maintenance). Disease education; medication routines; nutrition and physical activity; identification of personal goals; and barrier mapping with concrete strategies.Session 2: (T1, 3 months; focus: monitoring). Symptom perception and interpretation; home tracking (glycemia, SpO_2_, peak flow, and blood pressure as relevant); warning signs and thresholds; digital logs; and feedback on data quality.Session 3: (T2, 6 months; focus: management). Action plans for symptom exacerbations; problem solving; when/how to contact providers; aligning self-management actions with care plans; and relapse prevention.Session 4: (T3, 12 months; focus: reinforcement and consolidation). Review of progress; consolidation of habits; prevention of regression; update of personal action plans; and refreshing digital literacy.

All sessions are delivered on a GDPR-compliant platform with appointment reminders and secure authentication. Between visits, patients receive brief cues to action (SMS/email reminders) and links to concise educational resources aligned with international clinical guidance (e.g., GOLD for COPD; ADA standards for diabetes; and ESC guidance for CVD) to reinforce self-efficacy and adherence to the agreed goals.

Each session was systematically aligned with the Behaviour Change Technique Taxonomy (BCTTv1) to operationalize the theoretical constructs derived from Bandura’s Self-Efficacy Theory and the Health Belief Model. Specifically, the intervention includes techniques such as information about health consequences, goal setting (behavior), action planning, self-monitoring of behavior, feedback on performance, and problem solving, which correspond to motivational and cognitive determinants of self-care behaviors. Mapping BCTs to each theoretical construct supports transparency, replicability, and evaluation of which techniques are most effective in promoting sustained self-care across conditions.

### 2.6. Instruments and Data Collection

We use validated instruments to capture global and disease-specific self-care, HRQoL, and social support. Administration occurs at T0, T1, T2, and T3 through encrypted forms; the platform enforces completeness checks, range validations, and time stamps. Nurse assistance is available for technical or comprehension issues.

**Self-Care of Chronic Illness Inventory (SC-CII)**. The SC-CII operationalizes Riegel’s theory across maintenance, monitoring, management (and confidence when applicable) as a trans-diagnostic measure [23]. Items are Likert-type; domain and total scores are standardized to 0–100 (higher = better self-care). The primary endpoint is the T0 → T3 change in total score, with domain-level analyses a priori. An increase of ≥8 points is considered clinically meaningful, consistent with the validation literature. Psychometrics across languages and Italian samples are robust (Cronbach’s α usually ≥0.80, test–retest ICC ≥ 0.75).

**Self-Care of Heart Failure Index (SCHFI, v7.x)**. Disease-specific self-care in HF, providing maintenance, management, and confidence scores, is standardized to 0–100 [24]. Psychometric testing has shown good reliability (Cronbach’s α = 0.76–0.90 across domains) and construct validity in multiple international validations, including the Italian version. Test–retest reliability was satisfactory (ICC > 0.70). While thresholds vary by study, values ≥ 70 are often interpreted as adequate self-care.

**Self-Care of Diabetes Inventory (SCODI)**. Targets T2DM self-care behaviors spanning adherence, lifestyle, and glucose self-monitoring, aligned with Riegel’s domains [25]. Scores are transformed to 0–100. The instrument demonstrates good internal consistency (Cronbach’s α = 0.80–0.88) and factorial validity in the Italian population, with acceptable test–retest reliability (ICC > 0.75).

**Self-Care of COPD Inventory (SCCOPD)**. Measures BPCO self-care (maintenance/monitoring/management), including medication routines, symptom recognition, and timely action [26]. Scores are transformed to 0–100. The SCCOPD has shown strong psychometric properties with Cronbach’s α = 0.83–0.89 and adequate construct validity in both English and Italian validations.

**EQ-5D-5L**. A generic HRQoL measure across five dimensions with five severity levels, plus a 0–100 VAS [27]. The EQ-5D-5L demonstrates high test–retest reliability (ICC ≥ 0.80) and convergent validity with SF-36 physical and mental summary scores. The validated Italian version has shown comparable psychometric performance. Administration: 2–3 min.

**SF-36**. A multidimensional HRQoL profile with eight domains and Physical/Mental Component Summary (PCS/MCS) using norm-based scoring (mean 50, SD 10) [28]. Psychometric analyses have shown high internal consistency (Cronbach’s α typically >0.85 for all subscales) and strong test–retest reliability (ICC ≥ 0.80) in both general and chronic populations.

**Multidimensional Scale of Perceived Social Support (MSPSS)**. Assesses perceived social support from family, friends, and significant others (12 items; 1–7 Likert), providing domain and total means (higher = more support) [29]. The MSPSS demonstrates excellent reliability (Cronbach’s α = 0.88–0.94) and factorial stability across languages, including the Italian adaptation.

**Data capture, scoring, and quality controls**. The eCRF performs automated scoring (including 0–100 transformations), flags outliers (e.g., straight-lining, extreme completion times), and generates consistency reports. An audit trail logs user ID, date/time, and rationale for any data edits.

**Missing data (instrument-specific)**. We follow the instrument manuals. Where permitted, if ≤20% of the items in a subscale are missing, we apply person-mean imputation within that subscale; otherwise, the subscale is marked as missing and handled at the analysis level. We will document domain-specific missingness in the Results section of the full study report and address it using dataset-level multiple imputation, as described in the Section 2.8.

**Language versions and permissions**. We employ validated Italian versions where available. EQ-5D-5L is used under EuroQol terms; SF-36 will follow the owner’s license (or the RAND equivalent, if adopted) and will be explicitly stated in the submission; MSPSS is used with proper citation. All permissions will be secured before data collection and summarized in Ethics and Permissions.

### 2.7. Outcomes

**Primary outcome**. Change in SC-CII total (0–100) from T0 to T3 (12 months), with domain-specific analyses (maintenance, monitoring, and management). An ≥8-point increase is treated as clinically important.

**Secondary outcomes**. Change in disease-specific self-care (SCHFI/SCODI/SCCOPD), HRQoL (EQ-5D-5L index and VAS; SF-36 PCS/MCS), perceived social support (MSPSS), and engagement/satisfaction with the digital education (brief evaluation form anchored to established telehealth assessment domains).

### 2.8. Statistical Analysis

Analyses will be conducted in R studios version 4.4.0. (R Foundation for Statistical Computing, Vienna, Austria; https://www.r-project.org). We will summarize baseline characteristics using means (SD) or medians (IQR) and frequencies (%). Normality will be assessed with Shapiro–Wilk tests and Q–Q plots.

**Primary analysis**. Within-subject change over time will be evaluated using repeated-measures ANOVA (or Friedman for non-normal data) on SC-CII total and domains. When the omnibus test is significant, we will perform timepoint contrasts with Bonferroni-adjusted *p*-values. We will report mean differences with 95% confidence intervals and effect sizes (partial η^2^ and Cohen’s d for pairwise changes) [30].

**Secondary analyses**. Analogous repeated-measures models will be applied to disease-specific self-care, EQ-5D-5L, SF-36 (PCS/MCS), and MSPSS. Subgroup analyses will examine differences by diagnosis (COPD vs. T2DM vs. CVD) and baseline digital literacy strata (derived from a short digital-skills screener). Multivariate linear regression will explore predictors of 12-month improvement (ΔSC-CII total), including age, sex, education, diagnosis, baseline score, and digital engagement metrics (attendance, completion time, and message interactions). Collinearity will be checked via the variance inflation factors.

**Missing data**. We will evaluate missingness patterns with Little’s MCAR test. If data are missing at random, we will apply multiple imputation (m = 5–10 datasets), including all variables used in the analysis models. Sensitivity analyses will compare imputed vs. complete-case results to assess robustness.

**Reliability**. Internal consistency will be assessed at each timepoint using Cronbach’s α, with α ≥ 0.70 considered acceptable [31].

### 2.9. Monitoring and Quality Assurance

A study monitor will convene monthly with the clinical team to review recruitment, protocol adherence, data integrity, and adverse events (none expected given the educational, minimal-risk nature). A monitoring plan defines triggers for corrective actions, documentation of deviations, and timelines. A 10% session audit (Section 2.4, Training and Intervention Fidelity) will verify fidelity.

A structured process and impact evaluation will be conducted in parallel with the main study. The process evaluation will assess implementation fidelity, acceptability, and feasibility of the digital educational intervention using standardized checklists and feedback questionnaires completed by Family and Community Nurses (FCNs) and General Practitioners (GPs). Before participant recruitment, FCNs and GPs will have the opportunity to review the digital platform, session content, and materials through a pilot usability test and focus discussion to ensure content validity and contextual appropriateness. The impact evaluation will combine quantitative indicators (adherence rates, engagement metrics, dropout analysis) with qualitative feedback from participants and healthcare professionals to identify facilitators, barriers, and areas for improvement.

Participants who withdraw or drop out during the study will be contacted by the research team to document the primary reason for discontinuation (e.g., loss of interest, health deterioration, technical barriers, or other personal circumstances). A standardized short exit form will be administered by FCNs via phone or email to capture this information in a non-judgmental and voluntary manner. These data will be analyzed descriptively to identify potential barriers to participation and to inform future implementation strategies.

### 2.10. SWOT Analysis

To provide a structured appraisal of the PROSELF protocol, a SWOT (Strengths, Weaknesses, Opportunities, Threats) framework was developed considering both patient- and nurse-level perspectives (Table 1). This analysis supports the interpretability and generalizability of the study design by highlighting internal and external factors that may influence implementation and outcomes.

### 2.11. Ethical Considerations

The study adheres to the Declaration of Helsinki (2013) and the ICN Code of Ethics for Nurses (2021) and complies with EU GDPR (2016/679) [32,33,34]. At manuscript submission, ethics approval is pending at the Campania 2 Ethics Committee. No recruitment or data collection will start before approval is granted. All participants will provide digital informed consent via secure authentication, with explicit statements on voluntariness, withdrawal, confidentiality, and data handling.

Participants may withdraw from the study at any time without consequences for their ongoing care. Interventions may be discontinued or rescheduled in case of significant worsening of the clinical condition, hospitalization, or explicit request from the patient. Concomitant care (e.g., pharmacological therapy, physiotherapy, and nutritional counseling) will continue as usual and will not be restricted by the study protocol. No additional or experimental interventions will be introduced during participation.

### 2.12. Data Management and Confidentiality

Data are pseudonymized at entry (unique alphanumeric ID), transmitted, and stored under encryption on EU-hosted servers with role-based access control and password policies. A data dictionary and version-controlled codebook define variables, permitted values, and scoring rules. Backups occur on a scheduled basis; retention is 5 years after study close-out, followed by secure deletion. Any data sharing will occur only in an aggregated/anonymized form.

### 2.13. Study Registration

The protocol will be prospectively registered on ClinicalTrials.gov prior to enrollment, with all core fields completed (objectives, outcomes, timepoints, and analysis plan), in accordance with journal and ICMJE standards.

### 2.14. Study Flow and Timeline

Table 2 summarizes participant progression, intervention schedule, and data-collection timepoints. Eligible patients with COPD, T2DM, or CVD will be screened by Family and Community Nurses and enrolled after digital informed consent. Four assessments occur at baseline (T0), 3 months (T1), 6 months (T2), and 12 months (T3). Each assessment includes completion of the SC-CII and disease-specific self-care scales (SCHFI/SCODI/SCCOPD), HRQoL measures (EQ-5D-5L, SF-36), and the MSPSS.

The 12-month nurse-led digital educational intervention consists of four individualized tele-education sessions (T0–T3) and asynchronous follow-up materials between sessions. Data are stored in encrypted servers and monitored monthly for quality assurance.

Recruitment will be conducted over approximately nine months across all participating centers until the planned sample size (N = 180) is reached. Enrollment will close upon completion of the recruitment window or attainment of the target sample, whichever occurs first. To address potential loss to follow-up, participants will receive automated reminders and flexible scheduling options for each tele-education session and assessment. Follow-up completeness will be reviewed monthly, and participants lost to contact will be reached by telephone and email (up to three attempts) before being considered dropouts.

## 3. Expected Results

PROSELF is expected to demonstrate sustained improvements in global and domain-specific self-care at 12 months, with the most pronounced gains anticipated in maintenance (daily routines and adherence) and monitoring (awareness and tracking), while management gains will reflect improved decision-making and help-seeking. Extending the program to one year is designed to consolidate behaviors beyond short-term motivational spikes, allowing time for habit formation, troubleshooting, and relapse prevention. Beyond improvements in self-care behaviors, we also expect enhancements in health-related quality of life (HRQoL), as assessed through the EQ-5D-5L and SF-36 instruments, along with stronger perceived social support (MSPSS), mediated by enhanced self-efficacy—a key mechanism of change strengthened by mastery experiences, feedback, and credible encouragement within the nurse–patient relationship [15]. The intervention intentionally blends theoretical coherence (Riegel’s self-care model) with behavioral science (self-efficacy and perceived benefits/barriers) to move beyond information provision toward capability building and actionable plans. International evidence shows that telehealth and nurse-led models can improve adherence, symptom control, and satisfaction, particularly when integrated into routine care rather than delivered as standalone applications [9,10,11,12]. PROSELF adapts these insights to the Italian territorial context and the PNRR Mission 6 vision, operationalizing digital proximity: technology augments, rather than replaces, the human, relational, and educational core of nursing. If effective, the model could be scaled across Case della Comunità networks, informing standardized digital nursing pathways that bridge hospital, community, and home.

### Strengths and Limitations

A central strength of PROSELF is its theory-driven design: the intervention content, the primary outcome (SC-CII), and the analysis plan are aligned with the three domains of self-care, ensuring conceptual integrity and interpretability. The 12-month follow-up allows assessment of behavioral consolidation—often missing in shorter digital education trials. The protocol emphasizes fidelity (standardized training, session audits), data integrity (automated checks, audit trail), and methodological transparency (SPIRIT/STROBE), enhancing reproducibility. Limitations include the non-randomized sampling and reliance on self-reported measures, which may introduce selection and response biases. These risks are mitigated by standardized administration, validated instruments, sensitivity analyses, and subgroup exploration (e.g., by digital literacy). While adverse events are not expected, we remain vigilant to potential discomfort during telecommunication and provide technical support. The absence of a control group limits causal inference; however, the longitudinal design, domain-specific outcomes, and multivariate modeling support credible real-world evidence. Future research should consider multicenter randomized designs and cost-effectiveness analyses.

## 4. Discussion

This protocol outlines the rationale and methodological framework for evaluating a nurse-led, digitally supported educational program aimed at improving self-care and adherence among patients with chronic conditions. By grounding the intervention in established behavioral theories and integrating standardized instruments, PROSELF contributes to the advancement of evidence-based community nursing practice. Future research following implementation will assess the program’s effectiveness in real-world settings and its potential scalability within the national digital health strategy. Recent evidence reinforces the relevance of digitally supported nurse-led interventions for chronic disease management. Telemedicine approaches have been shown to significantly reduce hospitalizations and improve adherence in patients with heart failure and other chronic conditions [35,36]. Similarly, caregiver- and community-based digital education programs have demonstrated sustained behavioral improvements and better recovery outcomes across diverse contexts [37]. These findings further support the applicability and generalizability of the PROSELF framework within contemporary community and digital health settings.

## 5. Conclusions

PROSELF presents a comprehensive, nurse-led digital education model to strengthen self-care in chronic disease management. By explicitly operationalizing maintenance, monitoring, and management over 12 months, the study aims to deliver clinically meaningful, durable improvements in behavior and quality of life while advancing the role of nurses in digitally enabled community care. The protocol is fully compliant with ethical and reporting standards and aligned with national reforms under the PNRR Mission 6—Salute. Ultimately, PROSELF seeks not only to generate robust empirical evidence but to shape a scalable blueprint for European digital-health nursing, where human connection, technological innovation, and professional excellence come together to transform chronic care.

## Figures and Tables

**Table 1 healthcare-13-02972-t001:** SWOT analysis of the PROSELF protocol from patient and nurse perspectives.

Domain	Strengths	Weaknesses	Opportunities	Threats
Patient’s perspective	Personalized, theory-based tele-education addressing self-care behaviors. Accessible digital format, reducing geographical barriers. Integration of validated instruments (SC-CII, SCHFI, SCODI, and SCCOPD).	Digital literacy variability may limit engagement. Possible attrition over the 12-month follow-up.	Empowerment through continuous feedback and digital literacy. Potential scalability in community-based chronic care programs.	Risk of low adherence to scheduled video sessions. Potential connectivity issues in rural areas.
Nurse’s perspective	Standardized training ensuring intervention fidelity. Clear theoretical foundation (Riegel’s self-care theory).	Increased workload for FCNs/APNs in initial phase. Need for ongoing supervision.	Strengthening of advanced nursing role within PNRR Mission 6. Creation of a replicable tele-education model.	Institutional and logistical barriers to digital infrastructure adoption.

**Table 2 healthcare-13-02972-t002:** Participant progress, intervention program, and data collection timeline.

Timepoint	Key Activities	Instruments Administered	Outcomes Assessed
T0 (Baseline)	Eligibility screening; informed consent; first tele-education session	SC-CII; disease-specific scale; EQ-5D-5L; SF-36; MSPSS	Baseline self-care, HRQoL, support scores
T1 (3 months)	Second tele-education session; reinforcement materials	SC-CII; disease-specific scale; MSPSS	Interim change in self-care and support
T2 (6 months)	Third session; adherence monitoring review	All instruments	Mid-term improvement and engagement
T3 (12 months)	Final session; overall evaluation	All instruments	Long-term self-care and HRQoL outcomes

## Data Availability

The data presented in this study are available on request from the corresponding author due to privacy and ethical restrictions. As this manuscript describes a study protocol, no datasets have been generated or analyzed yet.

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
