# Peer review of "Digital Educational Intervention to Improve Adherence and Self-Care in Chronic Patients: A Prospective Study Protocol (PROSELF)"

_healthcare, 2025, doi:10.3390/healthcare13222972_

Round 1
Reviewer 1 Report
Comments and Suggestions for Authors
Thank you for the opportunity to review the study protocol entitled ”Digital Educational Intervention to Improve Adherence and Self-Care in Chronic Patients: A Prospective Study Protocol (PROSELF)”. The protocol is useful and the results are interesting to specialists.
To be suitable for publication in the Healthcare journal, please find here my comments:
- The Abstract should be rewritten to meet the journal's criteria (eliminate ethical approval, and expected outcomes should be replaced with Results). Expected outcomes could be the hypothesis of the research, in case the authors would like to formulate them.
- Keywords should cover the content of the study. Please include them all.
- The introduction should be developed, referring to the content of the research that should be based on the purpose of the protocol and to highlight the necessity of it.
- Inclusion/exclusion criteria - using self-declared data or medical records?
- A SWOT analysis would be useful for presenting the protocol results (both from the patient and the nurse side)
- In order to be seen as generalizable, please insert some statistical data that could support the results.
Author Response
Comments 1: The Abstract should be rewritten to meet the journal's criteria (eliminate ethical approval, and expected outcomes should be replaced with Results).
Response 1: The Abstract has been revised according to the journal’s formatting criteria. The reference to ethical approval was removed, and the “Expected Outcomes” section was replaced with a “Results” paragraph describing the anticipated findings and implications, as required for protocol papers in Healthcare.
Location: pp.1;Abstract;LL.41
Note: Revisions highlighted in the manuscript in yellow
Comments 2: Expected outcomes could be the hypothesis of the research, in case the authors would like to formulate them.
Response 2: Following the reviewer’s suggestion, a specific subsection titled “Research Hypotheses” was added at the end of the Introduction to clearly articulate the theoretical assumptions of the PROSELF study. Three hypotheses were formulated based on the Middle-Range Theory of Self-Care of Chronic Illness, linking digital nurse-led education with improved adherence, self-care, and quality of life outcomes.
Location: pp.3;Introduction;LL.134-143
Note: Revisions highlighted in the manuscript in yellow
Comments 3: Keywords should cover the content of the study. Please include them all.
Response 3: The keywords have been expanded to comprehensively reflect the scope and content of the study. The revised list includes methodological, clinical, and contextual terms (self-care, adherence, digital health, nurse-led intervention, study protocol, and community nursing), ensuring accurate indexing and discoverability across academic databases.
Location: pp.2;Abstract;LL.58-60
Note: Revisions highlighted in the manuscript in yellow
Comments 4: The introduction should be developed, referring to the content of the research that should be based on the purpose of the protocol and to highlight the necessity of it.
Response 4: The Introduction has been substantially expanded to better contextualize the study and justify the need for the PROSELF protocol. The revised section integrates recent epidemiological data, highlights the relevance of digital and nurse-led educational interventions, and explicitly links the study’s rationale to the objectives of the Italian PNRR Mission 6 framework and to the Middle-Range Theory of Self-Care of Chronic Illness.
Location: pp.2;Introduction;LL.70-85
Note: Revisions highlighted in the manuscript in yellow
Comments 5: Inclusion/exclusion criteria - using self-declared data or medical records?
Response 5: The Methods section “Setting and Participants” has been expanded to clarify how eligibility will be verified. Data for inclusion and exclusion criteria will be extracted from electronic medical records and confirmed by Family and Community Nurses or General Practitioners. Self-declared information will only be used for sociodemographic and lifestyle variables, ensuring clinical accuracy and reproducibility.
Location: pp.3;Setting and participants;LL.164-169
Note: Revisions highlighted in the manuscript in yellow
Comments 6: A SWOT analysis would be useful for presenting the protocol results (both from the patient and the nurse side)
Response 6: A new subsection entitled “SWOT Analysis” has been added before the Ethical Considerations section. This table summarizes the internal and external factors influencing the PROSELF protocol from both patient and nurse perspectives, enhancing the interpretability and generalizability of the study design.
Location: pp.6;SWOT Analysis;LL.352-359
Note: Revisions highlighted in the manuscript in yellow
Comments 7: In order to be seen as generalizable, please insert some statistical data that could support the results.
Response 7: To strengthen the generalizability of the study, additional statistical information has been incorporated in the “Sample Size Justification” section. The revised text clarifies that the proposed sample size (N = 180) provides adequate statistical power and aligns with comparable studies, supporting external validity and representativeness of the target population.
Location: pp.4;Sample size justification;LL.177-184
Note: Revisions highlighted in the manuscript in yellow

Reviewer 2 Report
Comments and Suggestions for Authors
This is an interesting proposal and the 12-month follow up should provide valuable information not previously available in other published studies on this topic. You are to be commended on including three very common chronic conditions as the other published material tends to focus on a single condition, making this study more robust.
METHODOLOGY: The following issues do not appear to have been addressed within the two frameworks cited :
SPIRIT 2013:
Criteria for discontinuing or modifying allocated interventions for a given trial participant ( e.g. participant request or improving/worsening disease)
Relevant concomitant care and interventions that are permitted or prohibited during the trial.
STROBE 2007:
Explain how loss to follow-up will be addressed .
In addition: How long will recruitment last? Is there a defined time period or does it continue until you reach your target sample size?
REFERENCES: Many of the references are very old. In fact, only 5/30 references are dated within the last 5 years. While there is a lack of published material in this area, I would suggest you look at the following papers which are more recent:
Ribeiro et al (2025) Effect of telemedicine interventions on heart failure hospitalizations: A randomized trial. Journal of the American Heart Association. 14(6). https://doi.org/10.1161/JAHA.124.036241
Georgios et al (2020) Rationale and design of a risk-guided strategy for reducing readmissions for acute decompensated heart failure: the Risk-HF study. ESC Heart Failure. 7(5):3151-3160. DOI:10.1002/ehf2.12897
Zhou et al (2019) Caregiver-delivered stroke rehabilitation in rural China: The RECOVER randomized controlled trial. Stroke. 50(7). https://doi.org/10.1161/STROKEAHA.118.021558
Author Response
Comments 1: The following issues do not appear to have been addressed within the two frameworks cited:
"SPIRIT 2013:
- Criteria for discontinuing or modifying allocated interventions for a given trial participant ( e.g. participant request or improving/worsening disease)
- Relevant concomitant care and interventions that are permitted or prohibited during the trial."
"STROBE 2007:
- Explain how loss to follow-up will be addressed .
- In addition: How long will recruitment last? Is there a defined time period or does it continue until you reach your target sample size?"
Response 1: We thank the reviewer for this important comment. The manuscript has been revised to explicitly address the missing SPIRIT and STROBE items. Under the “Ethical Considerations” section, we added a description of the criteria for discontinuing or modifying interventions and the management of concomitant care. The “Study Flow and Timeline” section was expanded to specify recruitment duration, loss-to-follow-up procedures, and the defined time frame for achieving the target sample size.
Location: pp.9;Ethical considerations;ll.367-372; pp.10;Study Flow and Timeline;LL.401-408
Note: Revisions are highlighted in green in the manuscript
Comments 2: "REFERENCES: Many of the references are very old. In fact, only 5/30 references are dated within the last 5 years. While there is a lack of published material in this area, I would suggest you look at the following papers which are more recent:
- Ribeiro et al (2025) Effect of telemedicine interventions on heart failure hospitalizations: A randomized trial. Journal of the American Heart Association. 14(6). https://doi.org/10.1161/JAHA.124.036241.
- Georgios et al (2020) Rationale and design of a risk-guided strategy for reducing readmissions for acute decompensated heart failure: the Risk-HF study. ESC Heart Failure. 7(5):3151-3160. DOI:10.1002/ehf2.12897.
- Zhou et al (2019) Caregiver-delivered stroke rehabilitation in rural China: The RECOVER randomized controlled trial. Stroke. 50(7). https://doi.org/10.1161/STROKEAHA.118.021558. "
Response 2: As suggested, we have updated the bibliography to include recent peer-reviewed studies published between 2019 and 2025. The new bibliographic references (Ribeiro et al., 2025; Georgios et al., 2020; Zhou et al., 2019) have been integrated into the Discussion section to strengthen the scientific relevance and timeliness of the manuscript, and integrated into the numbering sequence by converting the citation format to Vancouver style (31-33).
Location: "pp.11;Discussion;LL.457-464; pp.13;References;LL.552-560"
Note: Revisions are highlighted in green in the manuscript

Reviewer 3 Report
Comments and Suggestions for Authors
The authors present a theory-based, nurse-led digital education across individuals with COPD, T2DM and CVD – termed PROSELF. Overall, this is a compelling protocol – however, there are several comments I have before I can endorse this for publication.
Major comments
It is not explicitly clear from the protocol how PROSELF has been theory informed. For example, no description is offered of self-efficacy theory or the health belief model. Constructs have been stated, but what is the justification for their inclusion? Additionally, is the relevance of these constructs likely to differ between conditions (e.g., COPD, T2DM). It is unlikely that risk perception is needing to be targeted in those with COPD, but may be more relevant for CVD.
Have the authors considered also integrating and mapping to the behaviour change techniques taxonomy? This would significantly strengthen the protocol – and outcomes for the research, with those techniques employed potentially having a major impact on self-care/management.
Minor comments
There are minor issues with abbreviations throughout – some are presented in the Abstract but should still be included initially within the introduction. In other instances, some abbreviations are used without the full term provided first. Please review the manuscript to ensure consistency.
Please specify examples in lines 99-100. What specific cognitive or psychiatric conditions would preclude participation?
To strengthen the intervention – there could also be some acknowledgement or linking to the Behaviour Change Technique Taxonomy (BCTTv1) which would allow you to select specific BCTs to use within each session of your intervention to target the constructs association with self-efficacy theory and/or the health belief model (e.g., Information about health consequences, goal setting, action planning etc).
Please report psychometric properties for each of the questionnaires included – this is provided for some but not others. Please be consistent.
Are there any plans for process and impact evaluation? For example, with respect to process evaluation will FCNs and GPs have the opportunity to review the digital educational intervention first?
Will those who drop-out be followed up to indicate a reason why?
The discussion contains a random statement and doesn’t appear to have been edited? This is confusing – “Authors should discuss the results and how they can be interpreted from the perspective of previous studies and of the working hypotheses. The findings and their implications should be discussed in the broadest context possible. Future research directions may also be highlighted.” See lines 302-305.
Author Response
Comments 1: It is not explicitly clear from the protocol how PROSELF has been theory informed. For example, no description is offered of self-efficacy theory or the health belief model. Constructs have been stated, but what is the justification for their inclusion? Additionally, is the relevance of these constructs likely to differ between conditions (e.g., COPD, T2DM). It is unlikely that risk perception is needing to be targeted in those with COPD, but may be more relevant for CVD.
Response 1: The Introduction has been expanded to clarify how the PROSELF protocol was theory-informed. A detailed explanation of Bandura’s Self-Efficacy Theory and the Health Belief Model was added, including their conceptual relevance and differential application across conditions. This addition justifies their inclusion and strengthens the theoretical foundation of the study.
Location: pp.3;Introduction;LL.106-118
Note: Revisions are highlighted in blue in the manuscript
Comments 2: Have the authors considered also integrating and mapping to the behaviour change techniques taxonomy? This would significantly strengthen the protocol – and outcomes for the research, with those techniques employed potentially having a major impact on self-care/management.
Response 2: We thank the reviewer for this valuable suggestion. The manuscript has been revised to explicitly describe the integration of the Behaviour Change Techniques Taxonomy (BCTTv1). The taxonomy has been used to map the intervention components, ensuring alignment between the theoretical framework (self-efficacy and health belief models) and the specific behavior change mechanisms targeted in each session. This addition enhances reproducibility and strengthens the methodological rigor of the protocol.
Location: pp.5;Training and intervention fidelity;LL.104-201
Note: Revisions are highlighted in blue in the manuscript
Comments 3: There are minor issues with abbreviations throughout – some are presented in the Abstract but should still be included initially within the introduction. In other instances, some abbreviations are used without the full term provided first. Please review the manuscript to ensure consistency.
Response 3: All abbreviations in the text have been corrected and arranged by inserting the full name first and then the acronym in brackets.
Location: "pp.1;Abstract;LL.28; 34-39; pp.2;Introduction;LL.64-65; 77-78; 120; 122; 127-130; pp.4;Study design;LL.149; pp.4;Setting and participants;LL.163
Note: Revisions are highlighted in blue in the manuscript
Comments 4: Please specify examples in lines 99-100. What specific cognitive or psychiatric conditions would preclude participation?
Response 4: We thank the reviewer for the suggestion. The exclusion criteria have been clarified to include explicit examples of cognitive and psychiatric conditions that may preclude effective participation (e.g., moderate to severe dementia, schizophrenia spectrum disorders, major depressive disorder with psychotic features, and acute manic or psychotic episodes). This specification enhances clinical clarity and replicability.
Location: pp.4;Setting and participants;LL.157-161
Note: Revisions are highlighted in blue in the manuscript
Comments 5: To strengthen the intervention – there could also be some acknowledgement or linking to the Behaviour Change Technique Taxonomy (BCTTv1) which would allow you to select specific BCTs to use within each session of your intervention to target the constructs association with self-efficacy theory and/or the health belief model (e.g., Information about health consequences, goal setting, action planning etc).
Response 5: We thank the reviewer for this insightful suggestion. The manuscript has been revised to explicitly link the Behaviour Change Technique Taxonomy (BCTTv1) to the self-efficacy and health belief frameworks. Each educational session now specifies the BCTs employed (e.g., information about health consequences, goal setting, action planning, self-monitoring, feedback, problem solving), ensuring conceptual alignment and improving intervention replicability and evaluation.
Location: pp.5;Intervention;LL.228-236
Note: Revisions are highlighted in blue in the manuscript
Comments 6: Please report psychometric properties for each of the questionnaires included – this is provided for some but not others. Please be consistent.
Response 6: We thank the reviewer for the helpful comment. The “Instruments and Data Collection” section has been revised to consistently include the psychometric properties (Cronbach’s α, test–retest reliability, and validity evidence) for each instrument. This ensures methodological transparency and allows readers to appraise the robustness of all measures used in the study.
Location: pp.6;Instruments and data collection;LL.249-250; 252-255; 259-261; 264-266; 268-270; 274-277; 280-282
Note: Revisions are highlighted in blue in the manuscript
Comments 7: Are there any plans for process and impact evaluation? For example, with respect to process evaluation will FCNs and GPs have the opportunity to review the digital educational intervention first?
Response 7: We thank the reviewer for this valuable suggestion. The manuscript has been revised to include a process and impact evaluation framework. FCNs and GPs will review the digital intervention materials during a pilot phase to ensure content validity and contextual adaptation. Process indicators (fidelity, feasibility, acceptability) and impact measures (adherence, engagement, dropout analysis) will be collected throughout the study to evaluate both implementation quality and outcomes.
Location: pp.7;Monitoring and quality assurance;LL.335-344
Note: Revisions are highlighted in blue in the manuscript
Comments 8: Will those who drop-out be followed up to indicate a reason why?
Response 8: We thank the reviewer for the suggestion. The manuscript now specifies that all participants who discontinue participation will be contacted to document the main reason for drop-out using a standardized exit form administered by FCNs. This information will support interpretation of attrition data and help identify barriers to adherence for future interventions.
Location: pp.8;Monitoring and Quality Assurance;LL.345-350
Note: Revisions are highlighted in blue in the manuscript
Comments 9: The discussion contains a random statement and doesn’t appear to have been edited? This is confusing – “Authors should discuss the results and how they can be interpreted from the perspective of previous studies and of the working hypotheses. The findings and their implications should be discussed in the broadest context possible. Future research directions may also be highlighted.” See lines 302-305.
Response 9: We appreciate the reviewer’s observation. The template placeholder sentence at lines 302–305 has been removed, and the Discussion section has been revised to provide a concise synthesis of the study’s contribution and future implications. The updated paragraph now contextualizes the importance of the PROSELF protocol and highlights directions for subsequent research.
Location: pp.11;Discussion;LL.447-453
Note: Revisions are highlighted in blue in the manuscript
